# Impact of Pesticides on Cancer and Congenital Malformation: A Systematic Review

**DOI:** 10.3390/toxics10110676

**Published:** 2022-11-09

**Authors:** Viviane Serra Melanda, Maria Eduarda A. Galiciolli, Luíza S. Lima, Bonald C. Figueiredo, Cláudia S. Oliveira

**Affiliations:** 1Instituto de Pesquisa Pelé Pequeno Príncipe, Rua Silva Jardim, 1632, Curitiba 80250-060, PR, Brazil; 2Faculdades Pequeno Príncipe, Avenida Iguaçu, 333, Curitiba 80230-020, PR, Brazil

**Keywords:** congenital malformations, cancer, pesticides, environmental contaminants, birth defect, neoplasm

## Abstract

Pesticide exposure has deleterious effects on human health and development; however, no review has been conducted on human exposure to pesticides and the risk of congenital malformations and cancer in the same cohort. We systematically reviewed the evidence for this relationship following the Preferred Reporting Items for Systematic Reviews and Meta-Analysis guidelines. Four databases, namely, PubMed, Scopus, Cochrane Library, and BVS, were searched for studies deposited till July 2020 that examined the influence of pesticide exposure on congenital malformations and cancer outcomes in the same cohort. Seven studies were systematically included in this review. Among these, four were case–control studies, two were cross-sectional studies, and one was a longitudinal cohort study. The sources of contamination were food, water, or exposure during agricultural work. A link between the occurrence of cancer, congenital malformations, and exposure to pesticides was observed in most studies.

## 1. Introduction

The use of chemicals, including pesticides, by humans has caused evident side effects on human health and development. Prominent side effects that have gained special attention include premature birth, congenital malformations, learning disabilities, behavioral disorders, asthma, allergies, early puberty, diabetes, reduced fertility, and cancer [1,2].

The influence of environmental contaminants on human health is a matter of concern. It is estimated that 25–33% of diseases are caused by environmental pollutants, with 43% of this burden falling on children up to five years old. However, the absolute burden of this type of disease varies across continents. Countries with developed economies have a lower burden of illness because of decreased environmental risk factors [3]. Hence, despite being a global problem, environmental contamination is more severe in underdeveloped and developing countries than in developed countries [4].

Animal- or plant-based food production has always relied on the use of agricultural defensive substances, known as pesticides, that aim to control pests on crops. However, residues of these substances remain in the environment and in aliments such as fruits, vegetables, and derivatives. Consequently, pesticides enter and accumulate in the body causing acute and chronic toxic effects [5]. Evidence suggests that environmental contamination affects human health and increases morbidities. Environmental stressors penetrate external barriers during respiration, ingestion, and dermal penetration, via absorption by mucous membranes, or via the transplacental barrier in the fetus [6,7].

Environmental contamination factors represent the majority of those causing the total burden of disease and death worldwide. A large proportion of affected individuals are less than 5 years old, which supports the need for environmental management by public health agencies [8,9,10]. Studies have shown a relationship between exposure to environmental contaminants, cancer, and congenital malformations [5,11,12]. Evidence also suggests that cancer is more prevalent in children with congenital malformations indicating exposure to mutagenic environmental conditions [13,14]. Below, we describe in detail each topic of this systematic review.

### 1.1. Cancer

Reports on human cancer are available from the beginning of ancient history. Egyptian manuscripts dated 3000 B.C. have reported bone lesions in Egyptian mummies, suggestive of osteosarcoma [15]. Cancer is a set of diseases that can affect any part of the body. An error in the cell multiplication process gives rise to abnormal cells, which form new damaged cells that multiply and form tumor masses, which can be benign or malignant. Malignant tumor cells migrate to other organs and tissues to form new tumors and metastasize in adjacent areas [16,17].

Cancer knowledge has evolved throughout history. Development in different areas such as anatomy, physiology, epidemiology, biology, and chemistry has driven this evolution. Moreover, the rapid advancements in modern medicine and related technology have paved the way for the prevention, early diagnosis, and efficient treatment of cancer [15]. Nevertheless, cancer-related mortality is increasing over the years, as estimated by the Institute for Health Metrics and Evaluation (Figure 1).

Cancer is the second leading cause of death globally, after death from cardiovascular diseases [18]. In 2020, approximately 19.3 million new cases of cancer and almost 10 million cancer deaths were estimated [19]; by 2040, the global burden of cancer is predicted to reach 30.2 million cases [20]. According to the International Agency for Research on Cancer, the cancers with the highest incidence in the global population in 2020, regardless of sex and age, were (I) breast—2,261,419 cases; (II) lung—2,206,771 cases; and (III) colorectal—1,931,590 cases (Figure 2) [20]. The mortality was higher in patients with lung cancer (18%), colorectal cancer (9.4%), and liver cancer (8.3%). According to an analysis by the Global Cancer Observatory, one in four men and one in five women will develop cancer during their lifetime; of these, one in eight men and one in eleven women will have cancer as their main cause of death [20].

Knowledge of the epigenomic components in tumorigenesis has contributed to improving the prevention, detection, and treatment of cancer [21]. Epigenetic factors include biochemical and molecular mechanisms that influence genomic function and DNA methylation. Recent studies have indicated that epigenetic states inherited through generations can be altered by the environment [22]. Non-modifiable and modifiable epigenetic mechanisms such as environmental and lifestyle factors, which encompass nutrition, behavior, stress, physical activity, work habits, smoking, and alcohol consumption, have the potential to alter the extent of genetic risk favorably or unfavorably [23]. Several studies have reported that environmental influences including chemical exposure pose a risk of causing genomic and epigenomic instability in organisms [24,25,26]. Although few studies have addressed the development of childhood cancer and exposure to environmental pollution in children, evidence exists for a significantly increased risk for childhood leukemia, lymphomas, and central nervous system cancer in children exposed to environmental contaminants [27].

### 1.2. Congenital Malformations

Structural or functional abnormalities, including metabolic disorders, present at birth are defined as congenital malformations [28]. The set of diseases that comprise this category is included in Chapter XVII of the International Classification of Diseases [29].

Congenital malformation is a condition that can occur at any stage of embryonic development but is more likely to occur in the first trimester of pregnancy. Hence, the diagnosis may be made before, during, or after birth [30]. In addition to genetic factors, maternal exposure to teratogenous environmental factors, such as infectious diseases (rubella, syphilis, etc.), and the consumption of toxic substances (alcohol, solvents, pesticides, etc.), may be involved in the development of congenital malformations [27,31,32]. Most congenital malformations are not specifically associated with a single cause, but rather with the interaction of various genetic and environmental factors in a complex etiopathogenesis [33,34].

According to the Global Health Observatory data, the mortality in children aged 0–4 years was 5,642,669 in 2016. Among these, congenital malformation was the fifth cause of mortality, with 489,913 deaths, corresponding to 8.7% of all deaths [29]. According to the World Health Organization, approximately 303,000 children die annually from congenital malformation before turning one year old [34]. Between 3% and 4% of live births involve carriers of some type of congenital malformation [34]. According to the Centers for Disease Control and Prevention, deaths from congenital malformation were the primary cause of death in children under five years of age in the United States, followed by premature birth and external causes in 2018 [33]. Congenital malformations exacerbate infant mortality in a similar manner in European countries also [35]. Approximately 2.5% (104,000) of babies are born each year with some type of CM in Europe [34].

An estimated 3.2 million people survive with congenital malformations [32]. However, the infant mortality rates due to congenital malformations are believed to be underestimated owing to challenges faced in diagnosis during situations such as unidentified abortions and early infant deaths [35]. The World Health Organization has encouraged the implementation of congenital malformation surveillance programs to enhance the understanding of congenital malformation prevalence through improved data collection and analysis strategies [36].

Comparative studies between diagnostic sensitivity through prenatal examinations and postmortem analysis of congenital malformations indicate that there is a variation of approximately 3–9% due to disagreements related to false positives or false negatives. However, variations in the potential for prenatal detection depend on the type of congenital malformation, with the congenital malformations of the central nervous system, genetics, genitourinary system, and skeletal muscle showing the greatest concordance between prenatal diagnosis by ultrasound and postmortem analysis [37]. The more subtle the congenital malformation, the lower the prenatal diagnostic sensitivity [38,39,40,41].

Regarding the global distribution of congenital malformations, the prevalence indicator depends on the location-wise detection capacity which may lead to imprecise estimation of the global indicator [32]. Congenital malformation is a serious problem worldwide but is more impactful in low- and middle-income countries, where the disease rates range from 39.7 to 82 per 1000 live births. The African continent has the highest number of deaths caused by congenital malformations compared to other continents (Figure 3). However, when looking at the historical series (1990–2019) of deaths resulting from congenital malformations in the four continents (Africa, America, Asia, and Europe) whose data are available at the Institute for Health Metrics and Evaluation (IHME), it is characterized by a tendency to decrease the deaths by congenital malformations (Figure 3).

Congenital malformations have been associated with human exposure to chemical contaminants such as highly toxic pesticides [4]. The health hazards of environmental contamination including air pollution and contamination of water, soil, and food by chemical substances such as pesticides are more prominent in children under five years of age and cause neonatal health issues, neurodevelopmental diseases, endocrine disorders, and birth defects [7]. Therefore, it is strategic to combine innovative and multi-sectoral actions to understand and change the current scenario of congenital malformations and its impact on people’s lives and global public health. In this context, knowledge and warnings against current teratogenic exposures are essential for the prevention of congenital malformations.

### 1.3. Pesticides

Pesticides are chemical compounds used to eliminate pests (insects, rodents, fungi, and undesirable plants) that can harm agriculture and are vectors of diseases in humans. With the modernization of North American agriculture in the 1960s, pesticides were used at a large scale to increase productivity [42,43]. More than 1000 different types of pesticides are used worldwide [44].

To avoid undesirable losses in agriculture, the consumption of pesticides has been growing in recent years driven by population growth, lack of incentives due to crop uncertainties, and the absence of political actions. Currently, approximately 2 million tons of pesticides are used worldwide, with values ranging on average between USD 3 to USD 13/kg, representing an expanding market comprising 50% herbicides, 30% insecticides, and 20% of fungicides, rodenticides, and nematicides, that are being commercialized [37,45]. Moreover, the number of pesticides used per hectare of crops has been increasing worldwide (Figure 4).

The use of pesticides is regulated by agencies in each country where pesticides are marketed according to local regulations. The processing time for official registration and approval of use varies depending on the organizational conditions of each country, its evaluation and inspection capacity, and the influence of the agroeconomic sector. Some countries have a short process, while a few others can take more than a year between registration application and commercialization (Figure 5) [46,47].

Even though the importance of government legislation on pesticide licensing is well-known, only 45% of countries have rules to control the use of any type of pesticide; 34% have rules only for agricultural pesticides, and 19% of them have no rules to control this trade. The registration of pesticides involves dossiers with extensive datasets and laboratory and field tests, in addition to monitoring the use and management of registered pesticides [48]. It involves pre-registration, registration, and post-registration, in addition to reviewing existing records. Nevertheless, the process may differ from country to country depending on the legal and administrative organizations and the availability of human and financial resources in the country [46]. Although it is a complex process, 66% of countries have only a limited human capacity enrolled for pesticide registration, and 7% use outsourced services to carry out registrations [48].

Pesticide residues are not restricted to the places where they are applied; they are disseminated in the environment. Many pesticides can accumulate in the environment for years and, consequently, in the food chain causing adverse effects on the ecosystem and human health [44]. The consumption of pesticides is a concerning factor, as it is still high globally. The four main agricultural producers in the world are the United States, Brazil, China, and the European Union, and understandably, they are the largest users of agricultural pesticides [48]. However, attention should be paid to the amount of pesticide per growing area because it can be a better indicator in terms of accuracy. Saint Lucia is the leading country in terms of pesticide use per area of cropland (15%), followed by Japan (14%) and Colombia (13%) (Figure 6) [48].

Although European Union countries have banned the use of pesticides that pose serious risks to human health, they are allowed in the United States and several low-income countries [48,49]. Even though the European Union prohibits the use of pesticides that are very dangerous for human health and the environment, the export of many of these is not banned. Latin American countries, especially Brazil are the main destination for these exports [48]. Notably, banning one pesticide may lead to greater use of another of the same class, causing similar or larger risks. As prohibitions can increase the use of other substances, it is not clear to what extent these substitutions are favorable or whether it is an exchange of one harmful risk for another [49].

Faced with several factors that permeate international relations, the World Health Organization created the World Health Assembly Resolution 63.26 of 2010, which recognizes the disposal of large amounts of highly toxic pesticide residues, identifies the threat of unsafe storage of obsolete pesticides, and highlights the urgency of adopting and strengthening national policies and legislation aimed at eliminating risks to human health and the environment caused by pollution from obsolete pesticides. This resolution emphasizes the need to increase technical support and cooperation among world member states for the implementation of international conventions to reduce the adverse impacts of obsolete pesticides and chemicals [50].

Although pesticides contribute to food security, evidence shows that some of them are harmful to human health and the environment over the long term. In a large part of the world, agricultural labor does not have access to or is not properly trained in the use of PPE and handling of potentially toxic substances [47].

Studies involving pesticides and health bring an essential agenda to our society to contribute to a sustainable scenario in which the elementary need for humanity’s survival is primarily access to food for all. However, it does not exclude health care or the fight against illness generated by products that guarantee good agricultural performance but at the cost of toxicity.

### 1.4. Toxicology of Pesticides

The use of pesticides in crops is a practice that aims to protect production against harmful organisms but may cause collateral damage to the environment and human health [51]. Pesticides can be biological or chemical and can be of natural origin (alkaloid or non-alkaloid plant extracts) or synthetic (insecticides, rodenticides, fungicides). The mechanisms of pesticide toxicity involve two phases: kinetic and metabolic elimination [52]. Pesticides have different mechanisms of toxicity, with different toxicity potentials, which may vary according to the type of exposure, exposure duration, and type of active metabolite (Table 1) [53].

The parameters used for toxicological classification of pesticides include relative toxicity characters that adopt acute toxicological data (intraperitoneal, oral, dermal, and inhalation) and chronic toxicological data of short- and long-term toxicities on ocular and dermal lesions, dermal sensitization, neurotoxicity, carcinogenicity, and effects involving pre- and postnatal reproduction and development, in addition to genomic imprinting and endocrine disruption effects [52].

### 1.5. Rationale and Aims of the Systematic Review

Many studies have related human exposure to pesticides to the prevalence of diseases such as congenital malformations and cancer [5,13,14]. However, data on pesticide exposure and the consequent toxic effects are limited in estimating the overall incidence, prevalence, and death impacts of congenital malformations and cancer. Thus, in this systematic review, we aimed to approximate data related to human exposure to pesticides and the risk for manifestations of congenital malformations and cancer among the exposed population and their descendants

## 2. Materials and Methods

A systematic review is a study of the second type with a rigorous methodology, whose comprehensive process summarizes kinetic evidence from primary studies; therefore, it produces reliable evidence and auditable results [54]. This systematic review was conducted in accordance with the Preferred Reporting Items for Systematic Reviews and Meta-Analysis (PRISMA) guidelines [55]. The research question was elaborated from the acronym PECO, where P = population; E = exposure; C = comparison; O = outcome [56] that sought to detect, evaluate, and interpret all relevant research on “Is the prevalence of cancer and congenital malformation in the same population influenced by pesticides exposure?”.

Initially, the International Prospective Register of Ongoing Systematic Reviews (PROSPERO) was searched for studies that had already answered our research question or were in progress. Being unable to identify any compatible register, we registered our research protocol in PROSPERO (#CRD42020192534). As the work was a systematic review of the literature, approval from the Institutional Review Board was not required.

The descriptors were defined in English according to the Descriptors in Health Sciences; Boolean operators were applied to group the search terms, by using OR to expand the search for similar descriptors and AND to include all terms to be present in the same search. Thus, the search query was defined as “pesticide OR herbicide OR chloride OR organochlorine AND congenital anomaly OR congenital abnormalities OR congenital malformation OR congenital anomalies OR birth defect AND cancer OR neoplasm”.

The search process was performed in PubMed, Scopus, Cochrane Library, and BVS bases. We used the Start^®^ program (State of the Art through Systematic Review) from the planning stage to the extraction stage to elaborate on the protocol and organize the articles. The inclusion and exclusion criteria are listed in Table 2. Three reviewers (MEAG, VSM, and CSO) selected the relevant papers as described here. MEAG and VSM independently read the titles and abstracts of the papers and classified them as accepted or rejected. Subsequently, the accepted papers were read by both MEAG and VSM to which they applied the inclusion and exclusion criteria. In the extraction phase, MEAG and VSM scored the studies in terms of domains independently, without blinding the authors or publication status of the original studies, extracting the articles for systematic review. Cases of non-consensus in all stages were evaluated and defined by a third reviewer, CSO.

### Risk of Bias

The bias assessment was performed by ACROBAT-NRSI (A Cochrane Risk of Bias Assessment Tool for Non-Randomized Studies), a tool proposed by the Cochrane collaboration to assess the quality of observational cohort and control case studies for systematic intervention reviews because of its ability to assess causal relationships. This tool is composed of seven domains that evaluate pre- and post-intervention biases according to the following scales: low risk of bias, serious risk of bias, critical risk of bias, and no information [57].

## 3. Results

The systematic search identified 301 articles published in the four databases for this review. We excluded two duplicated studies and 287 papers that did not meet the eligibility criteria. Of the remaining twelve studies, two were excluded because we could not access the full text of the papers. After reading the remaining ten papers, three were excluded due to a lack of information. Thus, seven studies were finally included in our study (Figure 7).

The characteristics of the reviewed papers are summarized in Table 3. The seven papers selected for this study were from seven different countries: Germany, Russia, Norway, India, Brazil, the United States, and Italy. The evaluation period ranged between 1925 and 2010, but the papers were heterogeneously published between 1990 and 2015. Most studies were case–control studies (57%), followed by cross-sectional (28%) and cohort (15%) studies. In the case–control studies, 145,771 individuals comprised the exposed population, and 495,751 individuals comprised the control population. One of the cross-sectional studies included a population of 1151 individuals (52% men and 48% women), and in the second cross-sectional study, the number of evaluated individuals was not determined. For the cohort study, the population was divided into parental (131,243 males and 105,403 females) and filial generations (146,934 males and 139,541 females). Taken together, these results indicate that approximately 57% of the evaluated individuals were directly or indirectly exposed to pesticides. Further, 36% were female; 41% were male, and the sex was unknown for 23% of the participants.

Several pesticides such as chlorides (trichloroethylene, tetrachloroethylene (PCE), vinyl chloride, and dioxin), organochlorides (trans-1,2-dichloroethylene (DCE), β-hexachlorocyclohexane (β-HCH), γ-hexachlorocyclohexane (γ-HCH), Dichlorodiphenyldichloroethylene (p,p′-DDE), polychlorinated biphenyls (PCB), dieldrin, and endosulfan), and fungicides (mancozeb and benzene) were evaluated in the systematically selected studies. Contamination occurred mainly by food (40% of the studies), water (30%), or exposure to agricultural workers (30%).

The studies analyzed the occurrence of lung, throat, prostate, thyroid, breast, ovarian, and testicular cancers, along with that of other types of malignant tumors (neuroblastoma, Wilms tumor, soft-tissue sarcoma, malignant lymphoma, Hodgkin’s disease, coccygeal teratoma, Ewing’s sarcoma, and osteosarcoma). Furthermore, the studies linked the occurrence of cancer to congenital malformations such as neural tube defects, intestinal stenosis, renal and ureteral malformations, oral cleft defects, and reproductive tract birth defects. One study also analyzed the relationship between pesticides and fetal death.

### Risk of Bias

Most studies included in the present systematic review had a low risk of bias for all domains (Figure 8). Bias in confounding, measurement outcomes, and selection of reported results were uncertain in approximately 14% of the studies. None of the included studies had a high risk of bias.

## 4. Discussion

It is known that agricultural workers, workers applying pesticides, children, women of childbearing age, pregnant and lactating women, and the elderly are the individuals most susceptible to the effects of chronic pesticide exposure [63]. The risk of developing cancer is increased in individuals with birth defects; however, children with cancer and birth defects represent a small group. In addition, association studies can be challenging to be carried out [64]. The proximity of pesticide production areas to neighboring populations should be considered from the perspective of environmental impacts on human health [61]. Here, we conducted a systematic review aiming to investigate data related to human exposure to pesticides and the risk of manifesting congenital malformations and cancer in exposed people and their descendants.

The systematically selected studies used different tools and techniques to evaluate the possible relationship among pesticide exposure, cancer, and congenital malformation rates; for example, measurement of pesticide concentrations in tissues (fat tissue, blood, and milk) [57,58], questionnaire-based surveys to obtain health-related and socio-demographic data [27,60], analysis of medical records [31], and analysis of semen parameters [62]. These techniques are widely used because exposure to pesticides and their absorption can be measured through the biological monitoring of body fluids and tissues. Thus, these tissues are good markers because they provide a measure of the toxic effects of contaminants [65].

Some of the pesticides analyzed in the systematically reviewed papers were organo-chlorine, dioxin, mancozeb, and benzene. The Commission Implementation Regulation of the European Union determined that the approval of mancozeb use must not be renewed [66]. The commission recognized the substance as an endocrine disruptor for humans as well as other organisms. In addition, other pesticides such as DDT and dioxin have been banned by the United States Environmental Protection Agency since 1979. DDT and dioxin are considered carcinogenic and cause changes in the development of children by interfering with their reproductive system and hormonal profile [67].

Highly dangerous pesticides are registered or in use in 62% of countries in the world. Of these, only 41% of the countries assessed the need to review records due to the damage associated with these substances [68]. The population of Brazil consumes 20% of the total pesticides used worldwide, which represents 300 thousand tons per year. It is estimated that in the last 40 years, while the Brazilian agricultural area has increased only by 78%, the consumption of pesticides has increased by 700% [69]. In this context, Ibañez et al. [12] demonstrated a correlation between the prevalence of congenital malformations and crop productivity in the state of Paraná, Southern Brazil. Panis et al. [70] observed that pesticide contamination in drinking water increased the risk of cancer development in the Paraná, Southern Brazil.

Over the past few decades, studies have shown an association between pesticide exposure and changes in human health, both in exposed individuals and their offspring [27,71,72,73]. An individual is exposed to several sources of contact with pesticide-active metabolites that cause intoxication in the body. In this way, the same individual throughout his existence has contact with different toxic agents that can accumulate, interact, and potentiate their effects, causing damage to the body [74,75]. Farm workers are a category of people who are often exposed to pesticides; however, pesticides are also found in food and drinking water, contaminating the entire community [47,76]. Among the known consequences of pesticide poisoning, an increased risk of cancer and congenital malformations in offspring has been observed [77,78,79,80].

The articles analyzed in this systematic review differed from each other in several aspects, especially regarding the pesticides analyzed, in addition to the damage caused by such exposure. Thus, few studies have been conducted on these two factors (cancer and malformation) and exposure to pesticides in the same population [27,31,58,59,60,61,62]. The studies suggested further investigations to clarify the epidemiological conditions observed in relation to the risk caused by components in agricultural chemicals that cause potential contamination of the environment and harm to human health. Studies have indicated that a high degree of exposure may be linked to the increased prevalence of cancer and congenital malformations in exposed populations

## 5. Conclusions

In the review, we observed that the agricultural chemicals differed in their structure as well as their influence on human health regarding the rates of congenital malformation and cancer. Although more research is needed in this area, our systematic review indicates that there is a relationship between pesticide exposure, congenital malformations, and cancer rates in the same cohort. As far as the health-disease process is concerned, besides the objective relationship between the contaminant and its toxicity to human health, the sensitivity, and quality of the captured records, the capacity for toxicological analysis, factors related to living habits, working conditions, and sanitation are also discussed in the equation that aims to define the interface between the contaminant element, the contaminated object, and the subject’s illness. Knowledge of the impact of chemical exposure on human health reveals a complex perspective, with aspects that directly and indirectly affect the associations. According to the great diversity of pesticides, birth defects and neoplasms are manifested in various forms, and dispersion of similar groups occurs raising the need for prolonged temporal follow-ups, laboratory apparatus, and a team of professionals specialized in toxicological analysis for the studies to be reliable for scientific analysis.

## Figures and Tables

**Figure 1 toxics-10-00676-f001:**
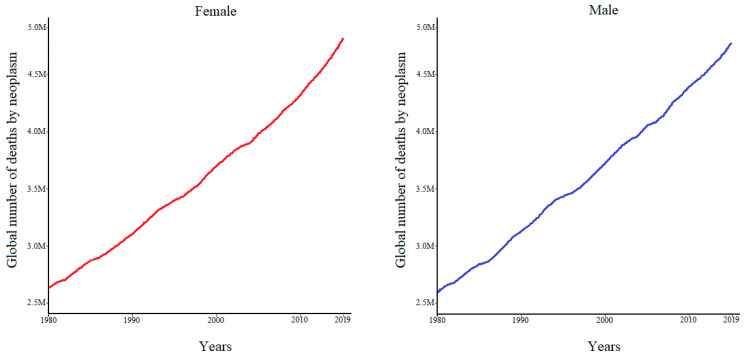
Global numbers of deaths by cancer over the years, according to the Institute for Health Metrics and Evaluation.

**Figure 2 toxics-10-00676-f002:**
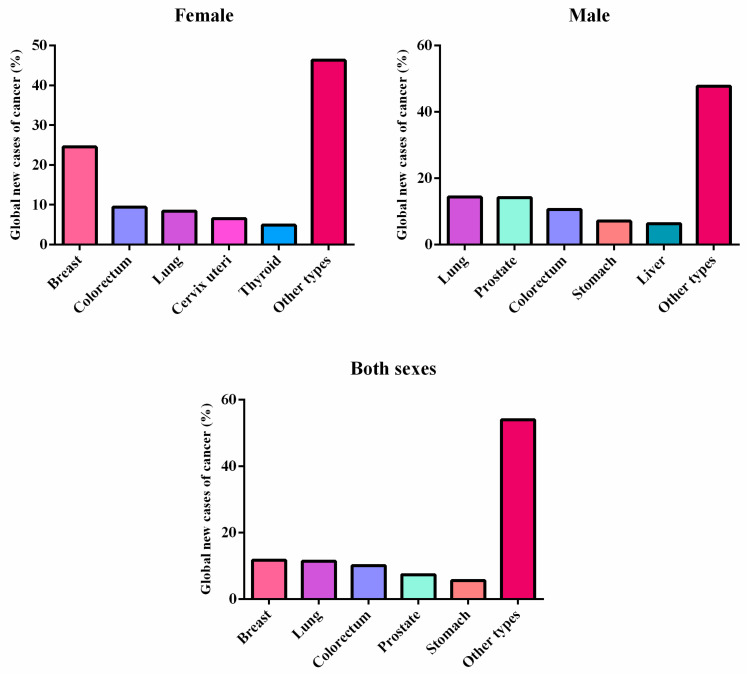
Incidence of global new cases of cancer (%) in 2020, according to the International Agency for Research on Cancer.

**Figure 3 toxics-10-00676-f003:**
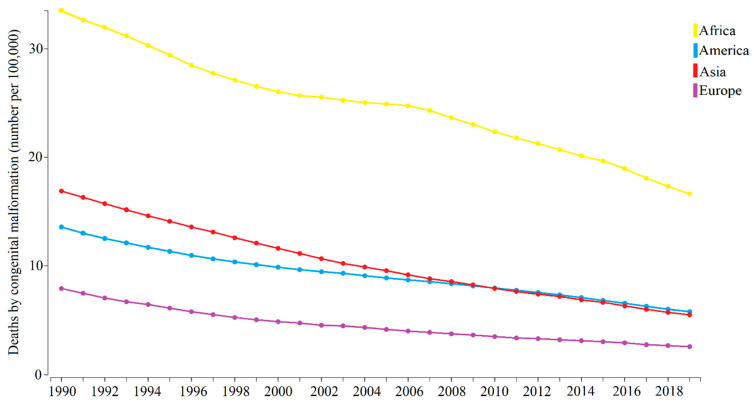
Deaths by congenital malformation over the years in Africa, America, Asia, and Europe, according to the Institute for Health Metrics and Evaluation.

**Figure 4 toxics-10-00676-f004:**
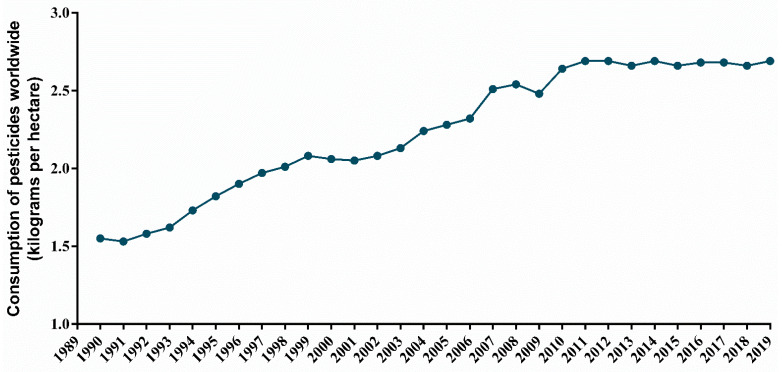
Kilograms of pesticides used by hectares of crops worldwide over the years, according to Food and Agriculture Organization of the United Nations.

**Figure 5 toxics-10-00676-f005:**
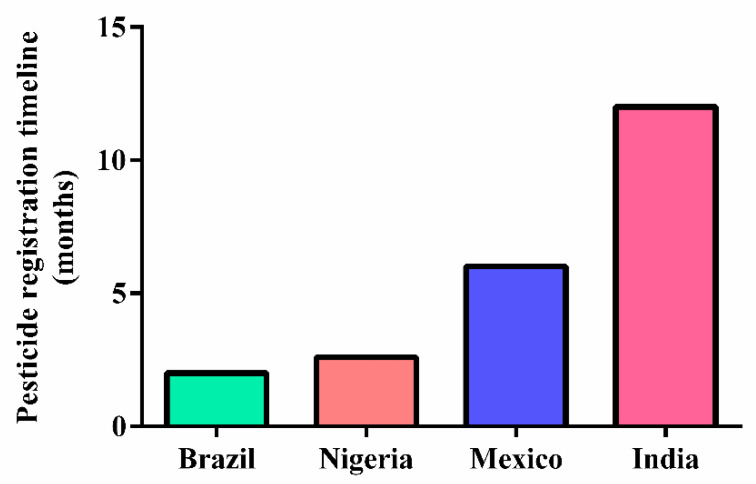
Pesticide registration timeline in different countries, according to World Health Organization.

**Figure 6 toxics-10-00676-f006:**
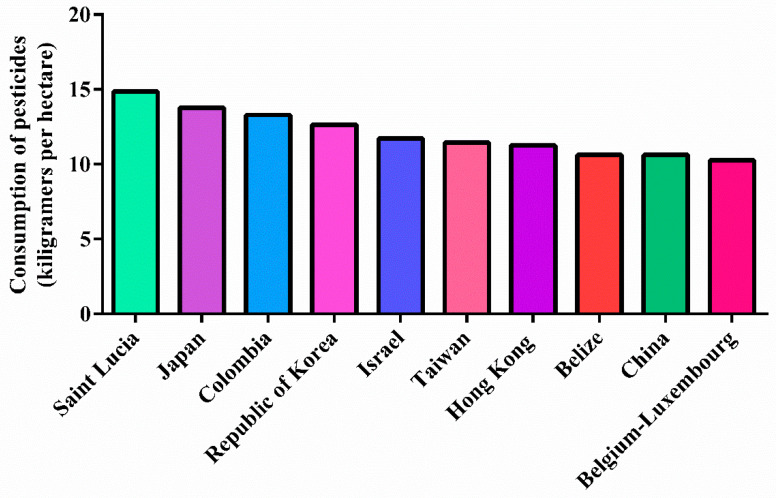
Kilograms of pesticides consumed by hectare in different counties, according to Food and Agriculture Organization of the United Nations.

**Figure 7 toxics-10-00676-f007:**
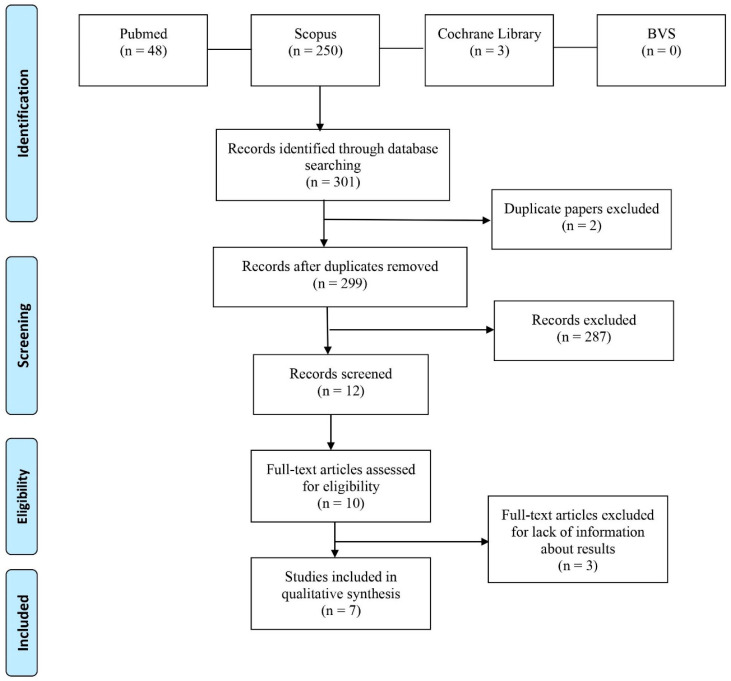
Prisma flow diagram.

**Figure 8 toxics-10-00676-f008:**
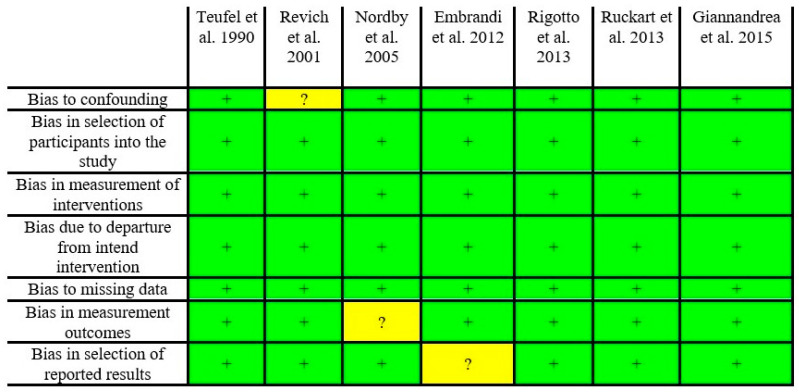
Risk of bias performed by ACROBAT-NRSI [57]. Green squares (+) mean a low risk of bias and yellow squares (?) mean an uncertain risk of bias [27,31,58,59,60,61,62].

**Table 1 toxics-10-00676-t001:** Mechanism of toxicity of pesticides.

Chemical Classification	Group	Mechanism of Toxicity	Effects on the Organism
Insecticides	Organochlorines	Prolongation of the opening time of sodium channels; competitive inhibitors of GABA-regulated chlorine flux; interference with cellular calcium by modification of membrane Ca^++^/Mg^++^ ATPase and calmodulin enzyme activity; organic bioaccumulation capacity; induction of liver enzymes.	Neurological disorders: paresthesia of the tongue, face and lips, tremors, mental confusion, coma, etc.; gastrointestinal changes such as: vomiting, abdominal colic, diarrhea, salivation, etc.; hepatic alterations: hepatic enzyme stimulation; carcinogenesis; mutagenesis; weight loss; general: anxiety, heart rhythm changes, arthralgia, memory loss, etc.
	Organophosphates	Blood cholinesterase inhibition.	Sialorrhea, tearing, nausea, vomiting, diarrhea, increased bronchial secretion, bradycardia, sweating, dyspnea, respiratory depression, miosis, hyperactivity, seizures, coma, and death.
	Pyrethroids	Leading to cholinergic crisis; inhibition of GABA-regulated chlorine flux; modification of membrane Ca^++^/Mg^++^ ATPase enzyme activity.	T syndrome: involuntary tremors; CS syndrome: choreoathetosis and salivation; excitatory neurotoxicity in: brain system, spinal cord, and peripheral nervous system; general: cutaneous, ocular, and gastrointestinal effects, convulsions, coma, respiratory paralysis and death.
Herbicides	Chlorophenoxyacetic acid	Intense lipid proliferation and depletion of the cofactor NADPH, causing cell death.	General: burning sensation in the mouth, nausea and vomiting, abdominal pain, diarrhea, dyspnea and hypoxemia, eye irritation, changes in level of consciousness, drowsiness, pulmonary edema, etc.
Rodenticides	Coumarins	Inhibition of the synthesis of vitamin K dependent factors; inhibition of prothrombin (factor II) synthesis in the liver; plasma thrombin precursors; vaso destructive action with capillary damage.	Hemorrhagic phenomena such as: epistaxis, purpura, petechiae, hematuria, and skin pallor.
	Ureics	Inhibition of complex I NADH ubiquinone reductase activity in mammalian mitochondria: complex I activity is related to insulin release in insulinoma cells and pancreatic islets. Affects the mitochondrial respiration of NAD-bound substrates in energy-demanding pancreatic islet cells.	General: dyspnea, cyanosis, pulmonary edema, increased bronchial and tracheal secretion, pleural effusion, etc.
	Fluorine compounds	Mimicking the action of acetate, incorporating it into the Krebs cycle. It turns into fluorine citrate; fluoride citrate is an inhibitor of aconitase and succinate dehydrogenase. Leading to blockage in the tricarboxylic acid cycle; blockade of tricarboxylic acids causes low energy production, leading to a reduction in oxygen consumption and cellular ATP concentration (mainly in cardiac and central nervous system cells).	Respiratory depression; cardiovascular system: hypotension, cardiac depression, ventricular tachycardia, fibrillation; general: nausea, vomiting, abdominal pain, agitation, anxiety, muscle spasm, stupor, reversible acute renal failure, coma, etc.
Fungicides	Heavy metal-based fungicides	The heavy metal binds to chemical groups such as carboxyl, amine, sulfhydryl, phosphoryl, imino, hydroxyl, interfering with cellular oxygen transport and energy production.	General: erythema, edema and vesicular eruptions, skin sensitization, nausea, vomiting, etc.; neurological: hyperexcitability or sedation.
	Phthalamides	Causes inhibition of Succinate Dehydrogenase Inhibitor (SDHI), interfering with mitochondrial function.	
	Triazoles	Activation of the key to oxidative stress and lipid peroxidation.	

**Table 2 toxics-10-00676-t002:** Inclusion and exclusion criteria.

INCLUSION	EXCLUSION
Incidence or prevalence of cancer and congenital malformations to contact with pesticides.	Absence of cancer and congenital malformation occurrence.
	The full paper was not found.
Relationship of environment contamination and cancer and congenital malformation.	The paper is not in English.
	Animal research.

**Table 3 toxics-10-00676-t003:** Summary of systematically reviewed studies.

Reference	Country	Study Period	Study Type	Sample Characteristics	Study Aim	Main Findings
Teufel et al. [58]	Germany	1985–1988	Case–control	Healthy children (n = 183) Children with malignant tumors (n = 46) Children with congenital malformations or benign tumors (n = 33)	To measure de chlorinated hydrocarbons in fat tissue of German pediatric patients.	The same chlorinated hydrocarbons (mainly, β-HCH and γ-HCH, dieldrin, p,p′-DDE, total PCB) were detected in all groups of pediatric patients, and no significant difference was observed in the increased concentrations of HCC in pediatric patients with malformations and tumors. However, the number of cases analyzed is still too small to rule out any possible risk.
Revich et al. [59]	Russia	1997–1998	Cross-sectional	Chapaevsk’s population	To measure the dioxin levels in milk and blood and to analyze data about cancer and congenital malformation of Chapaevsk’s population.	High levels of dioxin were detected in soil and water, as well as in women’s blood and milk. The rate of CMGC per child ranged from 0 to 10. The average number of CMGC per child was 4.5 for boys and 4.4 for girls.
Nordby et al. [31]	Norway	1925–1991	Cohort study	Parental generation Male (n = 131,243) Female (n = 105,403) Filial generation Male (n = 146,934) Female (n = 139,541)	To investigate the association of mancozeb exposure with cancer and congenital malformation in farmers’ families.	The study identified 131 cases of neural tube defect, of which 118 cases used pesticides in potato farming and 319 cases of thyroid cancer, of which 58 cases also used the same pesticide. The study concluded that neural tube defects were associated with potato farming, but thyroid cancer was not related to exposure to mancozeb.
Embrandiri et al. [60]	India	2008–2009	Cross-sectional	0–14 years old Male (n = 104) Female (n = 166) 15–30 years old Male (n = 209) Female (n = 210) 31–45 years old Male (n = 189) Female (n = 100) >46 years old Male (n = 96) Female (n = 77)	To verify the prevalence of common health problems associated with endosulfan in the Kasaragod district, India, seven years after the banning of this pesticide use in cashew plantation.	0–14 years old 46.0% of men and 42.5% of women had congenital abnormalities. 15–30 years old 30.4% of men and 31.7% of women had congenital abnormalities. 31–45 years old High prevalence rate of skin problems and infertility. >46 years old High prevalence rate of cancer and respiratory problems.
Rigotto et al. [61]	Brazil	2000–2010	Case–control	Case population habitants from three Brazilian cities known for their intensive use of pesticides (n = 145,509). Control Population Habitants from 11 other cities that do not use pesticides extensively (n = 494,939).	To compare morbidity and mortality related to pesticides between the case population and control population.	Increase in hospitalization rate due to cancer: case population, being 1.76 times higher than the control population. Annual increase in mortality rates due to neoplasia: 1.38 times higher in the case municipalities, compared to control. Fetal deaths: an increasing trend in case population and stability trend in the control population. Congenital malformation: there are no differences in rates between the case and control groups.
Ruckart et al. [27]	USA	1968–1985	Case–control	Children with cancer and/or congenital malformation (n = 58) Children without cancer and/or congenital malformation (n = 526)	To determine if children born during 1968–1985 to mothers with residential exposure to contaminated drinking water during pregnancy were more likely to have cancer and/or congenital malformation.	There was association between 1st trimester exposure to TCE and benzene and NTDs, and there is an association of response to monotonic exposure for TCE; results suggested weaker associations between 1st trimester exposure to PCE, vinyl chloride, and DCE and hematopoietic cancers in childhood.
Paoli et al. [62]	Italy	-	Case–control	Testicular cancer patients (n = 125) Control group (n = 103)	To investigate the possible role of occupational and environmental exposure to endocrine disruptors (polychlorinated biphenyls and hexachlorobenzene).	Semen analysis: sperm number and total motility were lower in cancer patients than in control patients and there were more malformations in cases samples than in controls. Organochlorine analysis: PCB congeners were found in 16 patients and no controls. HCB was detected in five cases and one control. The study also found a high association between higher levels of reproductive tract birth defects and an increased risk of testicular cancer.

β-HCH: β-hexachlorocyclohexane; γ-HCH: γ- hexachlorocyclohexane; p,p′-DDE: dichlorodiphenyldichloroethylene; PCB: polychtorinated biphenyls; TCE: trichloroethylene; NTD: neural tube defects; DCE: trans-1,2-dichloroethylene; PCE: tetrachloroethylene; CMGC: congenital morphogenetic condition.

## Data Availability

Not applicable.

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
