# Peer review of "Impact of Pesticides on Cancer and Congenital Malformation: A Systematic Review"

_toxics, 2022, doi:10.3390/toxics10110676_

Round 1
Reviewer 1 Report
The authors of the paper entitled: "Impact of pesticides on cancer and congenital malformation: systematic review" aimed to collect published papers that investigate the potential association between pesticide exposure and cancer development and congenital malformation in the human population. The basic idea of the proposed review is good, and the methodological description of the criteria chosen to include and/or exclude the published article on this topic is still satisfactory. However, there is a lack of interest in the toxicological properties of investigated pesticides which limited the scientific rigors of the proposed review. Why the authors did not put more emphasis on the toxicodynamic? Which are the toxicological mechanisms of proposed pesticides that led to the investigated adverse outcomes? Moreover, in the actual context of "mixture toxicology," the authors should at least mention the real problem of the concomitant exposure to different classes of pesticides and its potential association with human pathologies.
Reviewer 2 Report
The paper “Impact of pesticides on cancer and congenital malformation: systematic review“ is in the scope of the special issue Exposure, Bioaccumulation and Toxicity of Pesticides of the journal Toxics.
In general, an interesting and well performed review addressing relation among pesticides exposure, congenital malformation and cancer rates. This systematic review was carried out following the PRISMA guidelines and PROSPERO was searched for studies that had already answered the objective of research question or were in progress. Abstract states the major findings and conclusions of the paper. English language is appropriate and understandable. Cited references are relevant, and MS doesn’t include inappropriate self-citations. I suggest this paper to be published in Toxics after suitable revision.
Important: Image descriptions are missing!
Round 2
Reviewer 1 Report
-